

# Biochemical and genetic analyses of the oomycete *Pythium insidiosum* provide new insights into clinical identification and urease-based evolution of metabolism-related traits

Theerapong Krajaejun[1], Thidarat Rujirawat[2,3], Teerat Kanpanleuk[1], Pitak Santanirand[1], Tassanee Lohnoo[2], Wanta Yingyong[2], Yothin Kumsang[2], Pattarana Sae-Chew[2], Weerayuth Kittichotirat[4] and Preecha Patumcharoenpol[4]

[1] Department of Pathology, Ramathibodi Hospital, Mahidol University, Bangkok, Thailand
[2] Research Center, Faculty of Medicine, Ramathibodi Hospital, Mahidol University, Bangkok, Thailand
[3] Molecular Medicine Program, Multidisciplinary Unit, Faculty of Science, Mahidol University, Bangkok, Thailand
[4] Systems Biology and Bioinformatics Research Group, Pilot Plant Development and Training Institute, King Mongkut's University of Technology Thonburi, Bangkok, Thailand

Corresponding author
Theerapong Krajaejun,
mr_en@hotmail.com

## ABSTRACT

The oomycete microorganism, *Pythium insidiosum*, causes the life-threatening infectious condition, pythiosis, in humans and animals worldwide. Affected individuals typically endure surgical removal of the infected organ(s). Detection of *P. insidiosum* by the established microbiological, immunological, or molecular methods is not feasible in non-reference laboratories, resulting in delayed diagnosis. Biochemical assays have been used to characterize *P. insidiosum*, some of which could aid in the clinical identification of this organism. Although hydrolysis of maltose and sucrose has been proposed as the key biochemical feature useful in discriminating *P. insidiosum* from other oomycetes and fungi, this technique requires a more rigorous evaluation involving a wider selection of *P. insidiosum* strains. Here, we evaluated 10 routinely available biochemical assays for characterization of 26 *P. insidiosum* strains, isolated from different hosts and geographic origins. Initial assessment revealed diverse biochemical characteristics across the *P. insidiosum* strains tested. Failure to hydrolyze sugars is observed, especially in slow-growing strains. Because hydrolysis of maltose and sucrose varied among different strains, use of the biochemical assays for identification of *P. insidiosum* should be cautioned. The ability of *P. insidiosum* to hydrolyze urea is our focus, because this metabolic process relies on the enzyme urease, an important virulence factor of other pathogens. The ability to hydrolyze urea varied among *P. insidiosum* strains and was not associated with growth rates. Genome analyses demonstrated that urease- and urease accessory protein-encoding genes are present in both urea-hydrolyzing and non-urea-hydrolyzing strains of *P. insidiosum*. Urease genes are phylogenetically conserved in *P. insidiosum* and related oomycetes, while the presence of urease accessory protein-encoding genes is markedly diverse

in these organisms. In summary, we dissected biochemical characteristics and drew new insights into clinical identification and urease-related evolution of *P. insidiosum*.

## INTRODUCTION

Infectious diseases pose a greater threat to humans, animals, and plants as drug-resistant varieties emerge. Among these is pythiosis (the infectious condition caused by the fungus-like, highly invasive, oomycete microorganism *Pythium insidiosum*), which has been increasingly reported in tropical and subtropical countries (*Thianprasit, Chaiprasert & Imwidthaya, 1996*; *Krajaejun et al., 2006b*; *Gaastra et al., 2010*). Many healthcare personnel are not familiar with pythiosis. The use of anti-fungal drugs to control this pathogen has generally been ineffective (*Lerksuthirat et al., 2017*). Affected individuals often undergo surgical removal of the infected organ, and many succumb to the progressive disease (*Krajaejun et al., 2004*, *2006b*). Early and accurate diagnosis is necessary to ensure prompt and proper treatment, and thus an improved clinical outcome for patients. Isolation of the pathogen from infected tissues by the standard microbiological procedure is time-consuming and requires experience (*Chaiprasert et al., 1990*). A number of detection tools such as serological tests (*Pracharktam et al., 1991*; *Krajaejun et al., 2002*, *2006a*, *2009*; *Grooters et al., 2002*; *Jindayok et al., 2009*; *Supabandhu et al., 2009*; *Chareonsirisuthigul et al., 2013*; *Keeratijarut et al., 2013*; *Intaramat et al., 2016*), immunostaining assays (*Keeratijarut et al., 2009*; *Inkomlue et al., 2016*), and molecular biology methods (*Grooters & Gee, 2002*; *Botton et al., 2011*; *Keeratijarut et al., 2014*, *2015*; *Rujirawat et al., 2017*), have been successfully developed for *P. insidiosum* infection. However, such tools are not generally available in non-reference clinical laboratories, resulting in missed or delayed diagnosis of pythiosis.

Biochemical assays may be used to characterize *P. insidiosum* and could aid in the clinical identification of this organism. Different patterns of enzymatic activities in phosphatases, esterases, lipases, glucosidases, and proteases have been observed among strains of *P. insidiosum* (*Davis et al., 2006*; *Zanette et al., 2013*). Recently, Vilela and co-workers adopted an array of biochemical assays (hydrolysis of sugars, citrate, urea, esculin, etc.) to differentiate the pathogenic oomycetes, including six strains of *P. insidiosum* (*Vilela, Viswanathan & Mendoza, 2015*). They proposed that an ability to hydrolyze maltose and sucrose is a key biochemical feature to discriminate *P. insidiosum* from other mammalian-pathogenic oomycetes (i.e., *Lagenidium* species) and morphologically similar fungi. Although the use of these biochemical assays in the clinical identification of *P. insidiosum* is promising, it requires further evaluation with a more extensive selection of *P. insidiosum* strains.

In the current study, we evaluated 10 routinely available biochemical assays for characterization of 26 phylogenetically defined strains of *P. insidiosum*. The strains tested

had different geographic origins (i.e., Clade-I strains from Americas, Clade-II strains from Asia and Australia, and Clade-III mostly from Thailand) and were isolated from different hosts (i.e., humans and horses) (*Schurko et al., 2003*; *Chaiprasert et al., 2009*; *Rujirawat et al., 2017*). Initial assessment revealed strain to strain variation amongst the strains of *P. insidiosum* tested. The capacity to hydrolyze urea became our focus because this metabolic process relies on the enzyme urease, an important virulence factor of *Helicobacter pylori* and *Cryptococcus neoformans* (*Cox et al., 2000*; *Rutherford, 2014*; *Mora & Arioli, 2014*). Since the genome of *P. insidiosum* is publically available (*Rujirawat et al., 2015*), we were able to explore the genetic and evolutionary details of the urease gene in *P. insidiosum* and related oomycetes.

## MATERIALS AND METHODS

### Ethics statement

This study was approved by the Committee on Human Rights Related to Research Involving Human Subjects, at the Faculty of Medicine, Ramathibodi Hospital, Mahidol University (approval number ID 05-60-77).

### Microorganisms and growths

Twenty-six strains of *P. insidiosum* isolated from humans ($n = 14$) or equines ($n = 10$) with pythiosis and from the environment ($n = 2$), were available for this study (Table 1). Identity and genotyping (i.e., Clade-I, II, and III) of *P. insidiosum* were confirmed through culture identification, single nucleotide polymorphism-based multiplex PCR, and rDNA sequence analysis (*Chaiprasert et al., 1990, 2009*; *Badenoch et al., 2001*; *Rujirawat et al., 2017*). Because *P. insidiosum* has been classified as a Biosafety Level 2 organism (https://www.atcc.org), Biosafety Level 2 precautions were followed throughout this study (https://www.cdc.gov/biosafety). All of the organisms were retrieved from stock cultures, and maintained on Sabouraud dextrose (SD) agar at 37 °C for at least three passages. SD agar plugs (5 mm in diameter) from one-week-old, actively growing cultures of *P. insidiosum* were then prepared (*Krajaejun et al., 2010*; *Lerksuthirat et al., 2017*) for biochemical assays. Radial growth rate (mm/day) of *P. insidiosum* was evaluated, using the previously described method (*Krajaejun et al., 2010*; *Lerksuthirat et al., 2017*). Strains with growth rates $\geq 5$ mm/day were defined as fast-growing strains, while the rest were defined as slow-growing strains.

### Biochemical assays

To set up biochemical assays, 10 different routinely available agars were each prepared in test tubes (except the DNase assay agar, which was prepared in a Petri dish), using ingredients purchased from BD Difco and BBL (if not stated otherwise), and the recommended protocols of the manufacturers. These agars included: urea agar (urease assay), Simmons'citrate agar (citrate hydrolysis assay), bile esculin agar (esculin hydrolysis assay), DNA agar (DNase assay), and purple agar base (sugar hydrolysis assay) with 2% (wt/v) dextrose, lactose, maltose, sucrose (Merck, Darmstadt, Germany), trehalose

**Table 1 A list of 26 strains of *P. insidiosum* used for biochemical characterization in this study.**

| Strain ID | Reference strain ID | Source | Country | Phylogenetic clade | Growth rate (mm/day) | Fast/slow growth | Urease | Citrate | Bile esculin | Dextose | Lactose | Maltose | Sucrose | Trehalose | Xylose | DNase |
|---|---|---|---|---|---|---|---|---|---|---|---|---|---|---|---|---|
| Pi08 | CBS580.85 | Equine | Costa Rica | I | 10.6 | Fast | + | (−) | + | + | (−) | + | + | + | (−) | + |
| Pi03 | CBS577.85 | Equine | Costa Rica | I | 10.1 | Fast | + | (−) | + | + | (−) | + | (−) | + | (−) | + |
| ATCC28251 | ATCC28251 | Equine | Papua New Guinea | II | 9.5 | Fast | (−) | (−) | + | + | (−) | + | + | + | (−) | + |
| Pi10 | ATCC200269 | Human | USA | I | 9.0 | Fast | + | (−) | + | + | (−) | + | + | + | (−) | + |
| Pi02 | CBS579.85 | Equine | Costa Rica | I | 8.4 | Fast | + | (−) | + | + | (−) | + | + | + | (−) | + |
| Pi26 | N/A | Human | Thailand | II | 8.3 | Fast | + | (−) | + | + | (−) | + | + | + | (−) | + |
| Pi36 | ATCC64221 | Equine | Australia | II | 7.9 | Fast | + | (−) | + | + | (−) | + | + | + | (−) | + |
| Pi35 | Pi-S | Human | Thailand | II | 7.4 | Fast | + | (−) | + | + | (−) | + | + | + | (−) | + |
| Pi42 | CR02 | Environment | Thailand | II | 7.3 | Fast | + | (−) | + | + | (−) | + | + | + | (−) | + |
| Pi23 | N/A | Human | Thailand | II | 7.2 | Fast | (−) | (−) | + | + | (−) | + | + | + | (−) | + |
| Pi05 | CBS575.85 | Equine | Costa Rica | I | 7.0 | Fast | + | (−) | + | + | (−) | + | + | + | (−) | + |
| Pi09 | CBS101555 | Equine | Brazil | I | 6.6 | Fast | + | (−) | + | + | (−) | + | + | + | (−) | + |
| Pi51 | N/A | Environment | Thailand | III | 6.2 | Fast | (−) | (−) | + | + | (−) | + | + | + | (−) | + |
| Pi49 | N/A | Human | Thailand | III | 5.7 | Fast | (−) | (−) | + | + | (−) | + | + | + | (−) | + |
| Pi11 | N/A | Human | Thailand | II | 5.2 | Fast | + | (−) | + | + | (−) | + | + | + | (−) | + |
| Pi19 | N/A | Human | Thailand | II | 5.1 | Fast | + | (−) | + | + | (−) | + | + | + | (−) | + |
| Pi45 | MCC13 | Human | Thailand | III | 5.0 | Fast | (−) | (−) | + | + | (−) | + | + | + | (−) | + |
| Pi20 | CBS119455 | Human | Thailand | II | 4.6 | Slow | + | (−) | + | + | (−) | + | (−) | + | (−) | + |
| Pi50 | ATCC90586 | Human | USA | III | 4.2 | Slow | (−) | (−) | + | + | (−) | + | + | + | (−) | + |
| Pi07 | CBS573.85 | Equine | Costa Rica | I | 3.7 | Slow | + | (−) | + | + | (−) | + | + | + | (−) | + |
| Pi04 | CBS576.85 | Equine | Costa Rica | I | 3.7 | Slow | + | (−) | + | + | (−) | + | (−) | + | (−) | + |
| Pi46 | N/A | Human | Thailand | III | 2.6 | Slow | + | (−) | + | (−) | (−) | (−) | (−) | (−) | (−) | (−) |
| Pi47 | N/A | Human | Thailand | III | 2.4 | Slow | (−) | (−) | + | (−) | (−) | (−) | + | (−) | (−) | (−) |
| Pi44 | CBS119454 | Human | Thailand | III | 2.1 | Slow | + | (−) | + | (−) | (−) | (−) | (−) | (−) | (−) | + |
| Pi48 | N/A | Human | Thailand | III | 1.8 | Slow | + | (−) | + | + | (−) | + | + | + | (−) | + |
| CBS574.85 | CBS574.85 | Equine | Costa Rica | I | 0.7 | Slow | + | (−) | + | (−) | (−) | (−) | (−) | (−) | (−) | (−) |
| % Positive read ($n = 26$) | | | | | | | 73.1 | 0.0 | 100.0 | 84.6 | 0.0 | 84.6 | 76.9 | 84.6 | 0.0 | 88.5 |

**Notes:**
Information on strain identification numbers, sources of isolation, country of origins, assigned phylogenetic clades, rates of growth, and types of biochemical assays are provided in the table header. The symbol "+" and "(−)" indicate positive and negative biochemical reaction, respectively. Fast (≥5 mm/day) and slow (<5 mm/day) growths are determined based on mean radial growth rate. The strains CBS574.85 and ATCC28251 are included in this and other biochemical studies (*Vilela, Viswanathan & Mendoza, 2015*). The strains Pi07, Pi35, and Pi45 have their genome sequences available.

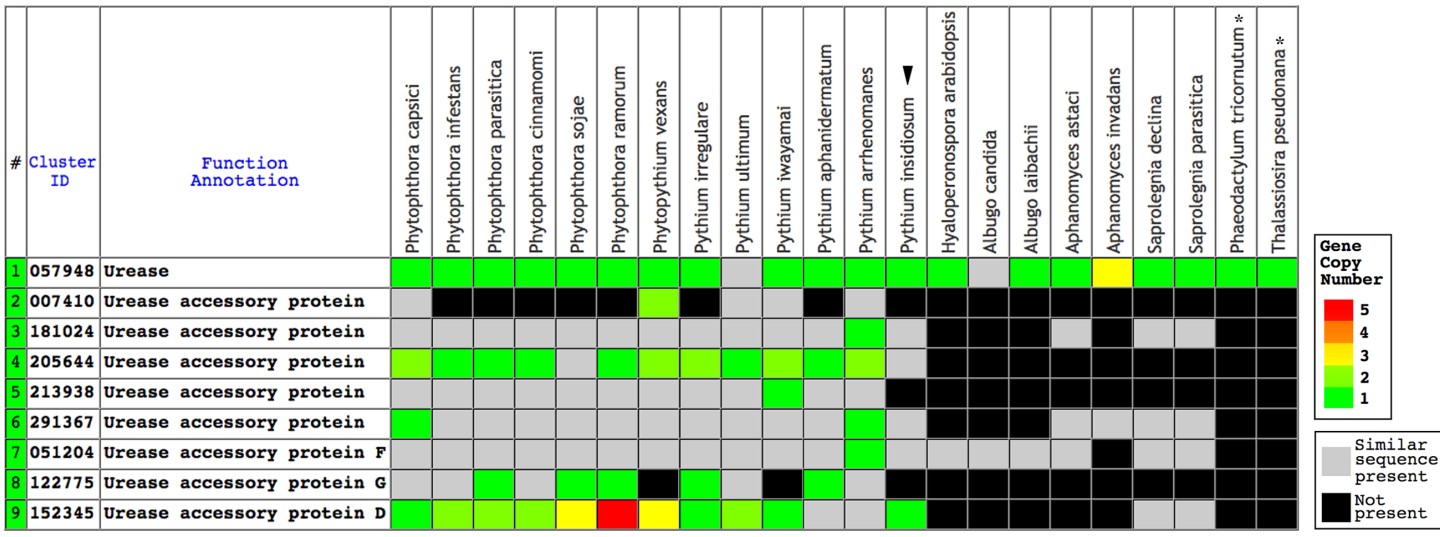

**Figure 1** The Oomycete Gene Table demonstrating the identified gene clusters containing the urease- and urease accessory protein-encoding genes presented in the genomes of *P. insidiosum* (arrow head), 19 related oomycetes, and two diatoms (asterisks). Cluster identification numbers (Cluster ID), function annotations, and identities of the genomes are shown in the table header. The arrow head indicates the genome of *P. insidiosum*. A gray box represents a similar sequence is identified, while a black box represents no similar sequence is found, in any given genome. Colored boxes refer to gene copy number.

(Sigma-Aldrich, St. Louis, MO, USA), or xylose. A 5 mm diameter agar plug of an actively growing colony of each *P. insidiosum* strain was placed upon each type of agar and incubated at 37 °C for two days before biochemical reactions were read. Each biochemical assay was interpreted as "negative" if the agar color remained unchanged, and interpreted as "positive" when the agar color changed: (i) from yellow to pink (urease assay); (ii) from brown to black (esculin hydrolysis assay); (iii) from green to blue (citrate hydrolysis assay); (iv) from dark blue to yellow (all sugar hydrolysis assays); and (v) from blue to colorless (DNase assay). All biochemical assays were performed in duplicate.

## Identification of urease- and urease accessory protein-encoding genes

The Oomycete Gene Table is an online comparative genomic analysis tool, derived from sequence similarity based gene grouping of the genome sequences of *P. insidiosum*, 19 related oomycetes, and two diatoms (Table S1) (*Kittichotirat et al., 2011*; *Rujirawat et al., 2018*). In the current study, the Oomycete Gene Table shows identification of putative urease- and urease accessory protein-encoding genes in the genomes of the oomycetes and diatoms (Fig. 1). Predicted urease protein sequences of the oomycetes and diatoms were aligned using MUSCLE (*Edgar, 2004*; *Dereeper et al., 2008*, *2010*), and assessed for sequence identity and similarity using NCBI BLAST (https://blast.ncbi.nlm.nih.gov/).

The urease and urease accessory protein sequences of the plant *Arabidopsis thaliana* (accession numbers: NP_176922 (urease structure protein, URE); NP_850239 (urease accessory protein D, URED); NP_173602 (urease accessory protein F, UREF);

and NP_180994 (urease accessory protein G, UREG)) (*Witte, Rosso & Romeis, 2005*) were retrieved from the NCBI database. To assess the presence of the orthologs in *P. insidiosum*, all of these *Arabidopsis* proteins were TBLASTN searched against the genome of the *P. insidiosum* strain Pi35 (also known as Pi-S), and two Illumina-derived genomes of the *P. insidiosum* strains Pi07 (also known as CBS 573.85) and Pi45 (*Rujirawat et al., 2015*; *Kittichotirat et al., 2017*; *Patumcharoenpol et al., 2018*), using the locally installed blast 2.2.28+ program (http://www.ncbi.nlm.nih.gov/) and the cut-off *E*-value $\leq 10^{-6}$.

## Phylogenetic analysis

Phylogenetic analysis of 24 urease-encoding sequences from *P. insidiosum* (strains Pi07, Pi35, and Pi45), related oomycetes, and diatoms (outgroup) (Table S1) was executed online at www.phylogeny.fr (*Dereeper et al., 2008*). In brief, the sequence alignment was performed by MUSCLE (*Edgar, 2004*). Poorly aligned positions or gaps were eliminated by Gblocks (*Castresana, 2000*). Phylogenetic relationships were calculated by PhyML, using the maximum-likelihood algorithm and the branch-assessing aLRT test (*Anisimova & Gascuel, 2006*; *Guindon et al., 2010*). The phylogenetic tree was reconstructed using TreeDyn (*Chevenet et al., 2006*).

## Sequence accession numbers

Sequences of the putative urease genes of *P. insidiosum* identified in the genomes of *P. insidiosum* strains Pi35 (accession number, LC317047 for *Ure*1), Pi07 (accession number LC325168 for *Ure*1), and Pi45 (LC325169 for *Ure*1A, and LC325170 for *Ure*1B) have been submitted to the DDBJ database.

# RESULTS

## Growth and biochemical characteristics of *P. insidiosum*

Twenty-six strains of *P. insidiosum* included in the current study were derived from different sources (humans, $n = 14$; animals, $n = 10$; and the environment, $n = 2$) and geographic origins (Asia, $n = 15$; Americas, $n = 10$; and Australia, $n = 1$). Based on the growth rates, *P. insidiosum* can be divided into two groups: (i) fast-growing strains (growth rate $\geq 5$ mm/day; $n = 17$; 65% of all strains), and (ii) slow-growing strains (growth rate $< 5$ mm/day; $n = 9$; 35% of all strains) (Table 1). Each group contained representatives from all phylogenetically distinct Clades (-I, -II, and -III), and from both humans and animals. Both environmental strains belonged to the fast-growing group.

As summarized in Table 1, all strains of *P. insidiosum* hydrolyzed esculin in the presence of bile but failed to breakdown citrate and two sugars (i.e., lactose and xylose). The majority of the strains can hydrolyze dextrose ($n = 22$; 85% of all strains), maltose ($n = 22$; 85%), sucrose ($n = 20$; 77%), trehalose ($n = 22$; 85%), and DNA ($n = 23$; 89%), while those that cannot utilize these substrates were almost all slow-growing. Unlike the other fast-growing strains, Pi03 did not hydrolyze sucrose. With regard to the urease assay, 71% ($n = 12$) of the fast-growing and 78% ($n = 7$) of the slow-growing strains could

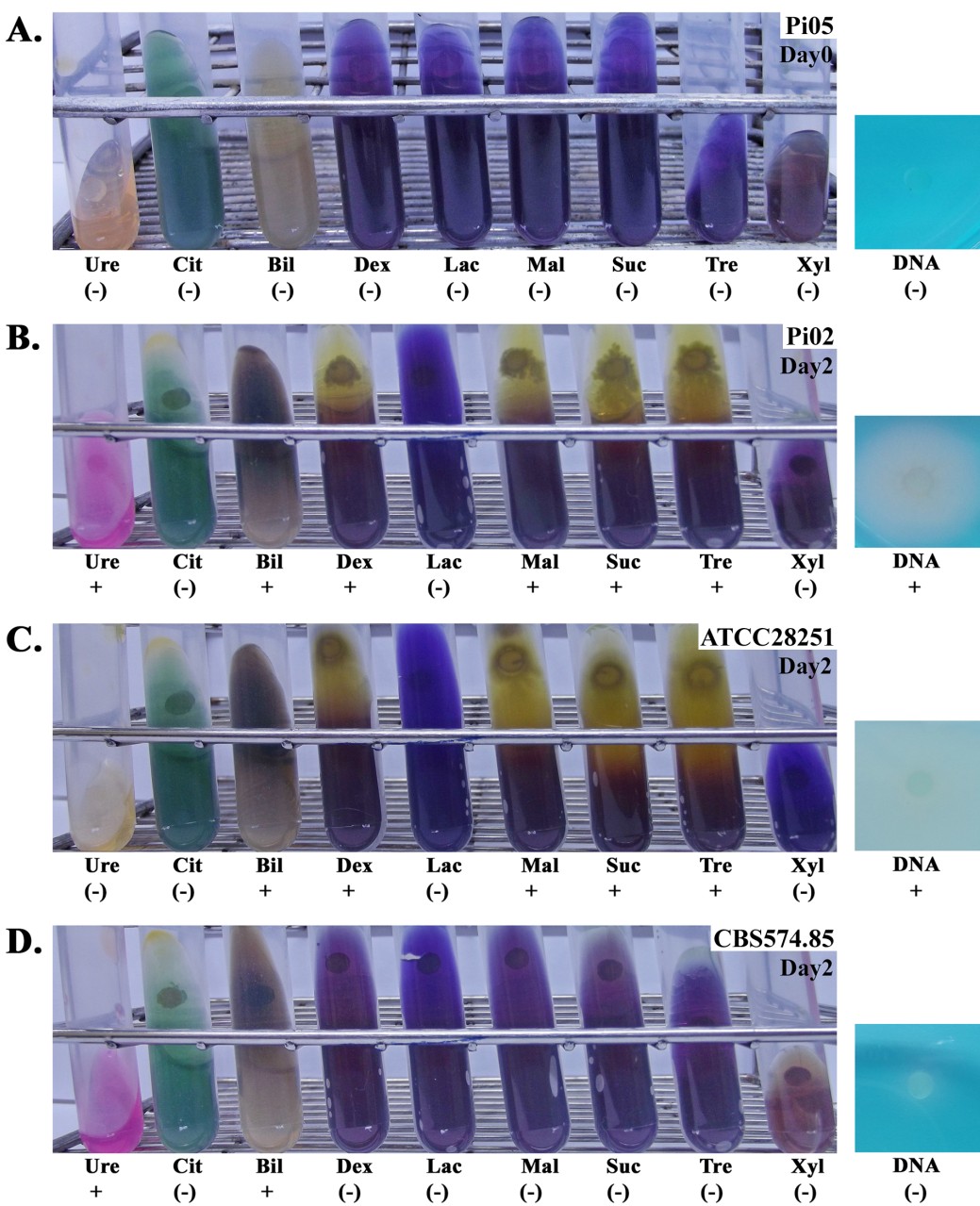

**Figure 2 Biochemical assays of four representative strains of _P. insidiosum._** Strain Pi05 (A) at the day of inoculation (Day#0; the colors of all agars remain unchanged), and strains Pi02 (B), ATCC 28251 (C), and CBS 574.85 (D) at two days post-inoculation (Day#2; biochemical results are read). Ten routinely available biochemical agars are included in this study: urea agar (Ure), Simmons'citrate agar (Cit), Bile esculin agar (Bil), DNA agar (DNA), and purple agar base with dextrose (Dex), lactose (Lac), maltose (Mal), sucrose (Suc), trehalose (Tre), or xylose (Xyl). The symbols "+" and "(−)" indicate positive and negative biochemical reaction, respectively. Photographs by Teerat Kanpanleuk.

catabolize urea. Biochemical characteristics of some representative strains at day 0 (all agar colors remained unchanged) and day 2 post-inoculation (all biochemical reactions were read) were displayed in Fig. 2.

## Ureases and urease accessory proteins of *P. insidiosum* and related oomycetes

Urease requires a number of urease accessory proteins to mediate enzymatic activity. Genes annotated as "urease" or "urease accessory protein" were searched using the Oomycete Gene Table (*Rujirawat et al., 2018*). All oomycetes and diatoms harbored a single copy of urease-encoding sequence (Gene cluster ID, #057948; average protein length: 849 amino acids; range: 761–1,345 amino acids), except the oomycete *A. invadans*, which contained three copies of this gene (Fig. 1; Table S1). Protein sequence alignment showed a high degree of identity (59–81%) and similarity (72–88%) between the ureases of oomycetes and diatoms (Fig. 3; Table S1).

A total of eight clusters of urease accessory protein-encoding genes were differentially presented in the genomes of 20 oomycetes (Fig. 1). These gene clusters included Cluster IDs: #051204 (found in 19 species), #291367 (17 species), #181024 (16 species), #152345 (15 species), #205644 (13 species), #213938 (12 species), #122775 (10 species), and #007410 (five species). Each oomycete genus possessed a different number of urease accessory gene clusters, for example: seven to eight clusters in *Phytophthora*, five to eight in *Pythium*, seven in *Phytopythium*, four in *Saprolegnia*, one to three in *Aphanomyces*, and one each in *Albugo* and *Hyaloperonospora*. None of these urease accessory gene clusters was identified in the diatom genomes.

TBLASTN search of the function-verified urease URE and urease accessory proteins URED, UREF, and UREG of the plant *A. thaliana* showed significant matches (*E*-value ≤−6) in the genomes of three representative *P. insidiosum* strains (Table 2): Pi07 (Clade-I strain), Pi35 (Clade-II strain), and Pi45 (Clade-III strain). One exception is UREF, which failed to find match in the genome of strain Pi07.

## Urease-based phylogenetic relationships

A set of 24 urease-encoding sequences identified in the genomes of *P. insidiosum*, related oomycetes, and diatoms (Fig. 1; Table S1), were subjected to reconstruction of a maximum likelihood-based phylogenetic tree. As expected, phylogenetic locations of the ureases of the diatoms (serving as an outgroup) were separated from that of the oomycetes. The oomycete ureases were allocated into three phylogenetically distinct clades (Fig. 4): (i) the clade of *Pythium*, *Phytophthora*, *Phytopythium*, and *Hyaloperonospora* species; (ii) the clade of *Aphanomyces* and *Saprolegnia* species; and (iii) the clade of *Albugo* species. Most of the organisms contain one copy of the urease-encoding gene, except *A. invadans* (three copies) and *P. insidiosum* strain Pi45 (two copies). Four urease-encoding sequences from the *P. insidiosum* strains Pi07, Pi35, and Pi45 were grouped together, and placed more proximally to non-*insidiosum Pythium*, *Phytophthora*, *Phytopythium*, and *Hyaloperonospora* species than to other oomycete species.

## DISCUSSION

A capacity to hydrolyze esculin, but not citrate, lactose and xylose, was the shared biochemical characteristic found in all 26 strains of *P. insidiosum* (Table 1), consistent with

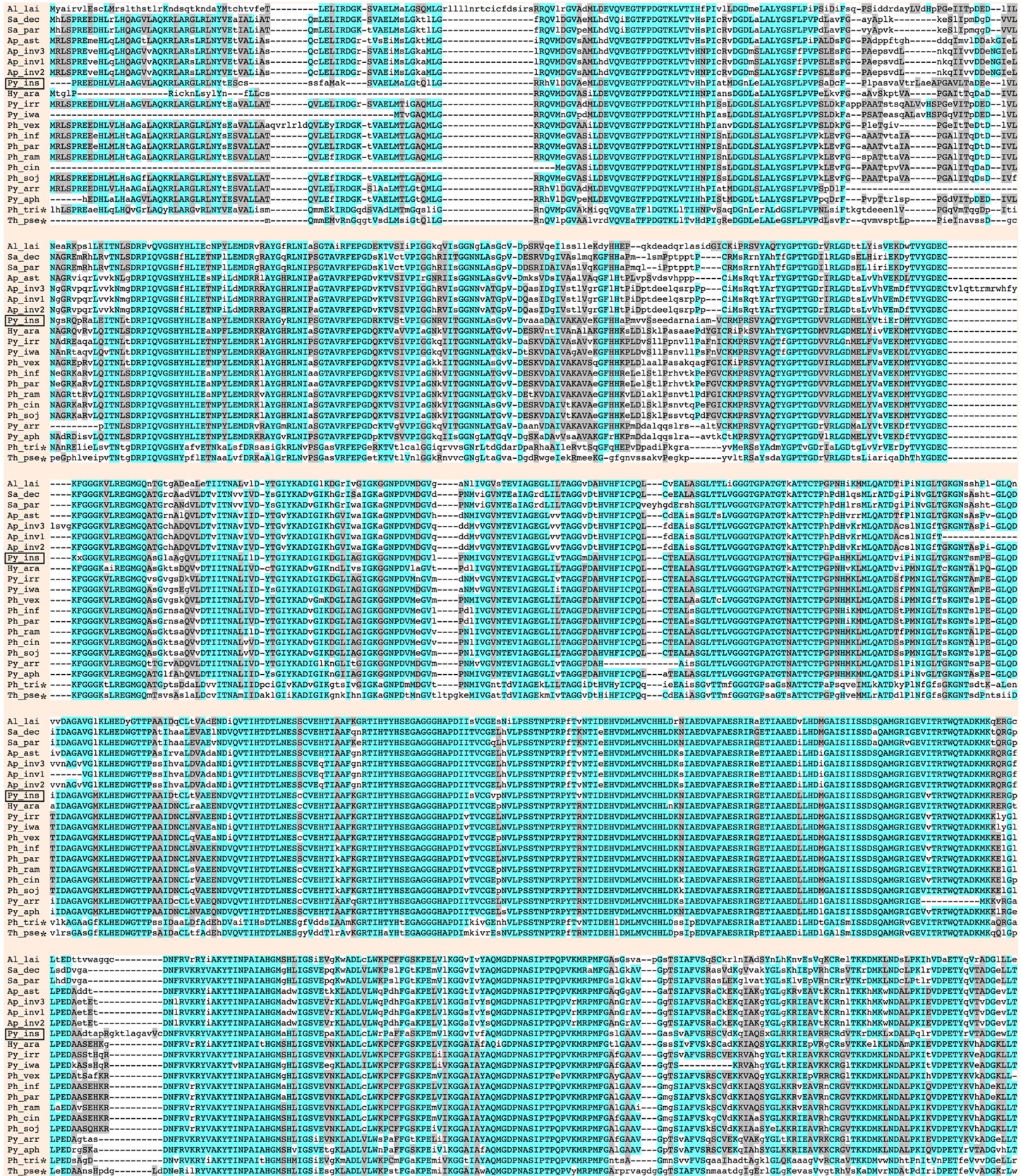

**Figure 3** Sequence alignment of full-length deduced urease proteins from *P. insidiosum*, related oomycetes, and diatoms. Initials of the genus and species names of each organism (Table S1) are listed on the left. The open box indicates *P. insidiosum*. The asterisks represent the diatoms. The symbol "-" indicates an absent amino acid in any given sequence. Cyan and gray colors highlight the identical and similar amino acids, respectively.

**Table 2** Urease and urease accessory protein orthologous sequences identified by TBLASTN search (cut-off *E*-value ≤−6) in the genomes of *P. insidiosum* strains Pi07, Pi35, and Pi45.

| Strain | Pi07 | Pi35 | Pi45 | |
|---|---|---|---|---|
| **Phylogenetic clade** | I | II | III | |
| **Growth rate (mm/day)** | 3.7 | 7.4 | 5.0 | |
| **Fast/slow growth** | Slow | Fast | Fast | |
| **Urease test** | + | + | (−) | |
| **Gene copy** | Copy-1 | Copy-1 | Copy-1 | Copy-2 |
| **URE** | | | | |
| *E*-value | 0.0 | 0.0 | 0.0 | 0.0 |
| Identity (%) | 64 | 63 | 64 | 64 |
| Similarity (%) | 75 | 73 | 75 | 76 |
| **URED** | | | | |
| *E*-value | 3E-19 | 2E-16 | 5E-26 | 2E-18 |
| Identity (%) | 42 | 42 | 31 | 47 |
| Similarity (%) | 60 | 58 | 52 | 67 |
| **UREF** | | | | |
| *E*-value | – | 1E-53 | 3E-54 | 6E-52 |
| Identity (%) | – | 42 | 42 | 40 |
| Similarity (%) | – | 61 | 61 | 59 |
| **UREG** | | | | |
| *E*-value | 6E-86 | 6E-60 | 5E-39 | 2E-33 |
| Identity (%) | 53 | 73 | 74 | 56 |
| Similarity (%) | 64 | 87 | 89 | 64 |

Notes:
The query sequences are the plant *A. thaliana* urease (URE; accession number, NP_176922) and urease accessory proteins D (URED; NP_850239), F (UREF; NP_850239), and G (UREG; NP_850239). Information on phylogenetic clades, growths, urease test results, gene copy, and TBLASTN search output (i.e., *E*-values, identity, and similarity) of *P. insidiosum* is summarized in the table.

the observations of *Vilela, Viswanathan & Mendoza (2015)*. The enzymatic components necessary to hydrolyze urea and certain sugars (i.e., dextrose, maltose, sucrose, and trehalose) were found in some strains but were not ubiquitous (Table 1). This finding contrasts with reports by Vilela et al., who showed all six *P. insidiosum* strains tested (including the strains CBS 574.85 and ATCC 28251 of the current study) could utilize urea and these sugars. This is especially important considering maltose and sucrose are two key sugars that were thought to differentiate *P. insidosum* from other pathogenic oomycetes and fungi (*Vilela, Viswanathan & Mendoza, 2015*). Failure to breakdown these sugars, in some strains, was markedly associated with slow-growth (growth rate, <5 mm/day) in *P. insidiosum* (Table 1). Because the biochemical characteristics varied among different strains (and even between different cultures of the same strain), caution is

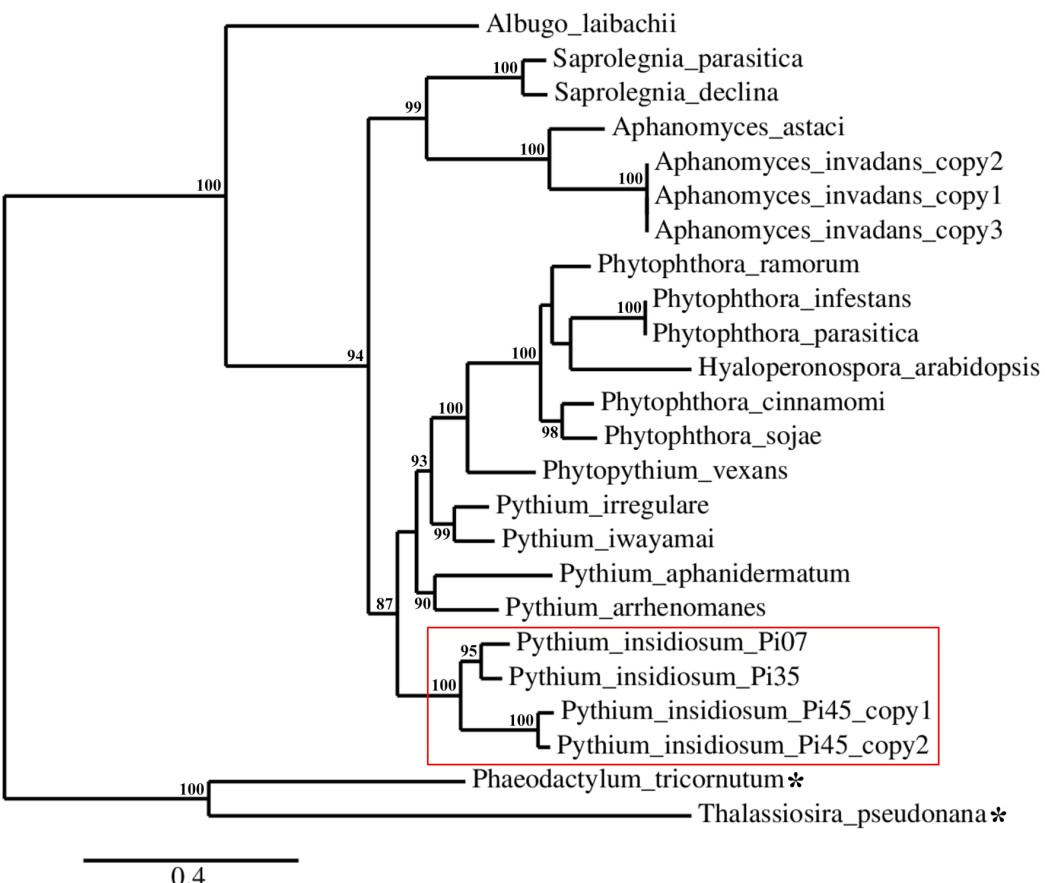

**Figure 4 Maximum-likelihood phylogenetic tree reconstructed from a set of 24 urease-encoding sequences identified in the genomes of *P. insidiosum*, related oomycetes, and diatoms (outgroup; as indicated by asterisks).** The oomycete ureases can be allocated in three phylogenetically distinct clades: (i) the clade of *Pythium*, *Phytophthora*, *Phytopythium*, and *Hyaloperonospora* species; (ii) the clade of *Aphanomyces* and *Saprolegnia* species; and (iii) the clade of *Albugo* species. Most of the organisms contain one copy of the urease-encoding gene, except *A. invadans* (three copies) and *P. insidiosum* strain Pi45 (two copies). The red box encompasses the urease sequences from three representative strains of *P. insidiosum*. Only branch support values ≥70% are shown at the nodes. The bottom bar reveals nucleotide substitution per site.

advised for the use of tests for the hydrolysis of maltose and sucrose in the clinical identification of *P. insidiosum* (especially for slow-growing strains).

Unlike the hydrolysis of sugars and DNA, the ability to utilize urea was not associated with growth rate in *P. insidiosum*. Efficient breakdown of urea can be observed in many slow-growing strains (i.e., Pi04, Pi07, Pi20, Pi44, Pi46, Pi48, and CBS 574.85), and not in all fast-growing strains (i.e., Pi23, Pi45, Pi49, Pi51, and ATCC 28251) (Table 1). The inability to utilize urea in a number of *P. insidiosum* strains could correspond to the lack of the urease-encoding gene, *Ure*1, in their genomes. We investigated the presence of *Ure*1 in the genomes of three representative strains of *P. insidiosum*, which included: (i) the urea-hydrolyzing, slow-growing, Clade-I strain Pi07; (ii) the urea-hydrolyzing, fast-growing, Clade-II strain Pi35; and (iii) the non-urea-hydrolyzing, fast-growing, Clade-III strain Pi45. All three strains contain *Ure*1 orthologous sequence, which significantly

matched the plant *Arabidopsis* urease (URE) (algorithm, TBLASTN; *E*-value, 0.0; identity, 63–64%; similarity, 73–76%; Table 2). Surprisingly, the non-urea-hydrolyzing strain Pi45 harbors two copies of *Ure*1 (designated as *Ure*1A and *Ure*1B), suggesting that the presence of *Ure*1 genes in the genome is not necessarily associated with the ability to hydrolyze urea in *P. insidiosum*.

In plants and microbes, urease accessory proteins (i.e., UreE, UreF, UreG, and UreD (orthologous to UreH)) are necessary for maturation and activation of the nickel-containing metalloenzyme urease (*Witte, Rosso & Romeis, 2005*; *Fong et al., 2013*). The urease structure protein (URE) and several accessory proteins (URED, UREF, and UREG) are required for enzymatic activity of the *Arabidopsis* urease (*Witte, Rosso & Romeis, 2005*). In addition to urease, we also sought evidence of urease accessory protein-encoding genes in *P. insidiosum*. TBLASTN search showed the URED, UREF, and UREG orthologs in the genomes of *P. insidiosum* strains Pi07, Pi35, and Pi45, as summarized in Table 2. A UREF ortholog was not found in the urea-hydrolyzing strain Pi07 (this may be due to the incompleteness of its genome), but URED and UREG orthologs were. Unlike the other strains, the non-urea-hydrolyzing strain Pi45 has two copies of both urease and urease accessory genes (Table 2). Since *P. insidiosum* generally contains a complete set of urease- and accessory protein-coding sequences, failure to utilize urea in some strains (Table 1) may be due to limited expression and/or down-regulation of these genes.

Genome analyses demonstrated that urease- and accessory protein-encoding genes are conserved in *P. insidiosum* from all three phylogenetically distinct clades, although gene duplication could occur in some strains (Table 2). We used the identified urease-encoding genes to further investigate metabolism-related evolution in *P. insidiosum*, non-human-pathogenic oomycetes, and diatoms (outgroup) (Table S1). The ureases are highly conserved in all organisms (Fig. 3), and their phylogenetic relationships are allocated as expected in the reconstructed tree (Fig. 4). However, the presence of urease accessory protein-encoding genes is diverse in these organisms (Fig. 1), ranging from: (i) harboring a wide variety of these genes in the genera *Phytophthora*, *Pythium*, and *Phytopythium*; to (ii) containing just a few genes in the genera *Hyaloperonospora*, *Albugo*, *Aphanomyces*, and *Saprolegnia*.

## CONCLUSIONS

No unique biochemical characteristic is observed among different strains of *P. insidiosum*, cautioning the use of related biochemical assays for pathogen identification. Unlike the hydrolysis of sugars, the ability to hydrolyze urea was not associated with *P. insidiosum* growth, as many slow-growing strains, and not all fast-growing strains, can utilize urea, even though the urease- and accessory protein-encoding genes are present and highly conserved in both urea-hydrolyzing and non-hydrolyzing strains of *P. insidiosum*. Future investigations on expression and regulation of the urease and accessory protein-encoding genes could elaborate the urea metabolism and its potential role in pathogenicity in *P. insidiosum*. Gain and loss of urease and accessory protein-encoding genes occurred in

the genomes of oomycetes and diatoms as their evolutions diverged. In the current study, we dissected several biochemical characteristics, and provided new insights into urease-based evolution of *P. insidiosum*.

## ACKNOWLEDGEMENTS

We thank Dr. Tristan Brandhorst and Dr. Thomas D. Sullivan for reviewing the manuscript and making valuable suggestions.

### Funding
This work is supported by the Faculty of Medicine, Ramathibodi Hospital, Mahidol University and the Thailand Research Fund (Grant number, BRG5980009). The funders had no role in study design, data collection and analysis, decision to publish, or preparation of the manuscript.

### Grant Disclosures
The following grant information was disclosed by the authors:
Faculty of Medicine, Ramathibodi Hospital, Mahidol University and the Thailand Research Fund: Grant number, BRG5980009.

### Competing Interests
The authors declare that they have no competing interests.

### Author Contributions
- Theerapong Krajaejun conceived and designed the experiments, performed the experiments, analyzed the data, contributed reagents/materials/analysis tools, prepared figures and/or tables, authored or reviewed drafts of the paper, approved the final draft.
- Thidarat Rujirawat performed the experiments, analyzed the data, prepared figures and/or tables, authored or reviewed drafts of the paper, approved the final draft.
- Teerat Kanpanleuk performed the experiments, prepared figures and/or tables, approved the final draft.
- Pitak Santanirand conceived and designed the experiments, contributed reagents/materials/analysis tools, approved the final draft.
- Tassanee Lohnoo performed the experiments, approved the final draft.
- Wanta Yingyong performed the experiments, approved the final draft.
- Yothin Kumsang performed the experiments, approved the final draft.
- Pattarana Sae-Chew performed the experiments, approved the final draft.
- Weerayuth Kittichotirat performed the experiments, analyzed the data, contributed reagents/materials/analysis tools, approved the final draft.
- Preecha Patumcharoenpol performed the experiments, approved the final draft.
## Ethics

The following information was supplied relating to ethical approvals (i.e., approving body and any reference numbers):

This study was approved by the Committee on Human Rights Related to Research Involving Human Subjects, at the Faculty of Medicine, Ramathibodi Hospital, Mahidol University (approval number ID 05-60-77).

## DNA Deposition

The following information was supplied regarding the deposition of DNA sequences:

Sequences of the putative urease genes of *P. insidiosum* identified in the genomes of *P. insidiosum* strains Pi-S (accession number, LC317047 for *Ure*1), Pi07 (accession number LC325168 for *Ure*1), and Pi45 (LC325169 for *Ure*1A, and LC325170 for *Ure*1B) have been submitted to the DDBJ database.

## Data Availability

The raw data are included in the 'Results' section as well as in the tables and figures in the article.

## Supplemental Information

Supplemental information for this article can be found online at http://dx.doi.org/10.7717/peerj.4821#supplemental-information.

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
