# Peer review of "Biochemical and genetic analyses of the oomycete Pythium insidiosum provide new insights into clinical identification and urease-based evolution of metabolism-related traits"

_PeerJ, doi:10.7717/peerj.4821_

## Round 0.1 · original submission · Major Revisions

We have received two helpful reports. Reviewer #2 calls your attention to the need for more information regarding the preparation of the cultures and confirmation that cultures were properly revived (especially regarding your failure to replicate the reported (Vilela et al.) growth of CBS 574.85 on maltose and sucrose ). In contrast to this reviewer, I do not think that the manuscript should be more narrowly focused on urease but I agree with him on the relative paucity of data on pathogeneses, and probable advantages of removing "pathogenicity" from the title.

Additional review level comments by the editor:
- I think a reference (and explanation) regarding the differences between clades-I -II and -III would help the reader.

- Since P. insidiosum is pathogenic, it is likely that you took special precautions in handling it during the experimental procedures. I think those precautions should be detailed in the methods section.

- There seems do be a discrepancy in the labeling of the strains ATCC 28251, which is called Pi37 in this paper ( Pi21 in Vilela et al.), and CBS 574.85, which is called Pi06 here but Pi19 in Vilela et al. The lack of a consistent strain numbering hampers the comparison between your data and those in the rest of the literature.

- I am not completely convinced about your conclusion that the different urease results suggest that they "might have evolved divergently to fit the particular lifestyles and niches of each organism. " Without additional data to confirm such evolutionary pressures, I do not see how your data cannot equally be explained by neutral drift, or why a similar argument cannot be invoked to claim "divergent adaptation" for each and every trait which happens to be found to be different between strains.

Reviewer 1 ·

Basic reporting

1. Basic Reporting
The article was written in a clear way, and sound technically correct. The manuscript is well structured and the references are in agreement with the objective of this study.

Experimental design

2. Experimental design
The research question is clearly postulated at the end of the introduction. This issue is relevant since the mechanisms of pathogenicity and metabolic pathways of this microorganism are not yet completely known.
The investigation was conducted rigorously and the methods used are in line with those prevalent in the field. The methods were well described and could be replicated by other researchers.

Validity of the findings

3. Validity of the Findings
The data presented in this manuscript highlighted the need of a more comprehensive knowledge of the biochemical features of this oomycete before to propose these features for identification and differentiation from others pathogens. The results of the biochemical profile analysis are similar to data already published by other authors, but the analysis of the urease and urease accessory protein sequences are new data and can be useful for future studies.
The data are robust, well explained on discussion and appropriately concluded.

Comments for the author

I have only minor correction to improve the manuscript:
Line 65: the authors cited Chaiprasert et al., 1990 to say that isolation of the pathogen from infected tissues is difficult. This statement is not very correct, since the isolation of the pathogen from infected tissue is easily done, at least from infected horses and experimental animals. Please, rephrase this sentence.
Line 75: “ in the clinical identification…”
Line 83: “Although the use…”
Line 114: “ ten, instead of 10”
Line 466, 468: Figure 1 legend – arrowhead, instead arrow head

Reviewer 2 ·

Basic reporting

The authors reported the phylogenetic tree and evolution of urease enzyme and provide the new information for the study of Pythium insidiosum.

1. Title: The experiment mainly worked on the urease in terms of phylogenetic tree and evolution. Thus, the appropriate of this tiltle should focus on more specific to include the key word of “Urease”. Also very less pathogenesis was mentioned in the discussion part thus the title should be considered. Other biochemical tests were like adding without much association of urease story. It might be better to consider for process another manuscript.

2. Keywords: No keywords couldnot be found.

Experimental design

3. Materials & Methods :
a. Microorganism & Growth : P 3 :
i. L. 109: One-week-old, actively growing colonies of P. insidiosum : It seems reasonable to prepare the inoculum like that for growth rate experiment. It is not clear that how the authors prepared the cultures which stocked before. These cultures require, at least, subculture 2-3 times to revive. This point will be very critical influence to the growth rate experiment.
ii. L. 112: It will be better to add reference for preparing the inoculum size : 5 mm.

Validity of the findings

4. Result :
a. L. 168: Sample: Please state whether the sample size was statistic calculation or …….. .
b. L. 174: The authors recruited strains from 3 Clades of Pythium insidiosum, from this study, what the reader can learn from the clade point of view.

5. Discussion :
a. L. 233: Failure to utilize the sugars was markedly associated with slow growth : What is the theory supported this result. Or in other hand, how do you prove that it is not the "co-incidence"?
b. L. 238: Utilization of urea not associated with growth rate : any mechanism or Similar as above comment. How do you know that it is not the co-incidence between the hydrolysis of sugar and growth rate? : Urea breakdown can be observed in slow growing strains : Normally, the newly isolates from host show more rapid growth that the old ones. How do know that the slow growing is not the effect from reviving the stock before study?

6. Conclusion : The main story should be very interesting if the manuscript is focused on urease story. Another notice is that the conclusion of "several biochemical characteristics, and provided new insights into pathogenicity and metabolism-related evolution of P. insidiosum.“ , isn’t it too broad as comments above.

Comments for the author

The authors presented and analysed of urease gene including phylogenetic tree and evolution using Pythium insidiosum strains from Thailand and from other regions. The manuscript will be more focused if the authors presented the story of urease.

---

## Round 0.2 · accepted · Accept

I think you have successfully addressed all queries.

#